# Latent Ergonomics Maps: Real-Time Visualization of Estimated Ergonomics of Human Movements

**DOI:** 10.3390/s22113981

**Published:** 2022-05-24

**Authors:** Lorenzo Vianello, Waldez Gomes, Freek Stulp, Alexis Aubry, Pauline Maurice, Serena Ivaldi

**Affiliations:** 1Université de Lorraine, CNRS, Inria, LORIA, F-54000 Nancy, France; pauline.maurice@loria.fr (P.M.); serena.ivaldi@inria.fr (S.I.); 2Université de Lorraine, CNRS, CRAN, F-54000 Nancy, France; alexis.aubry@univ-lorraine.fr; 3CIAMS, Université Paris-Saclay, F-91405 Orsay, France; 4Department of Cognitive Robotics, Institute of Robotics and Mechatronics, German Aerospace Center (DLR), 82234 Wessling, Germany; freek.stulp@dlr.de

**Keywords:** ergonomics tools, digital human model, wearable sensors

## Abstract

Improving the ergonomy of working environments is essential to reducing work-related musculo-skeletal disorders. We consider real-time ergonomic feedback a key technology for achieving such improvements. To this end, we present supportive tools for online evaluation and visualization of strenuous efforts and postures of a worker, also when physically interacting with a robot. A digital human model is used to estimate human kinematics and dynamics and visualize non-ergonomic joint angles, based on the on-line data acquired from a wearable motion tracking device.

## 1. Introduction

Poor ergonomics conditions in work environments may lead to serious work-related musculoskeletal disorders (WMSDs), including severe disabilities [1]. The development of WMSDs is an issue not only for the workers’ health and well-being but also represents an important cost for companies and society [2,3]. In recent years, there has been a surge in robotic solutions for ergonomics interventions, notably using industrial manipulators conceived for collaboration with humans (i.e., cobots) and exoskeletons [4,5]. These robotics solutions require ergonomics specialists to identify dangerous conditions and develop adequate interventions, whilst maintaining operational safety and productivity.

Classic kinematics ergonomics evaluation tools such as RULA, REBA and OWAS [6,7,8] use human joint positions to produce an ergonomics score for a given body posture. Kinematics-based scores are fast to compute, but dynamics aspects of the task may not be negligible [9]. Dynamics estimation may be an important complementary evaluation to the classic tools throughout an entire task execution, as they are more suitable for evaluating more accurately varied body morphologies, and external wrenches applied to the human body. Many recent works evaluate dynamic aspects of the task execution, such as internal and external human wrenches [10,11,12,13].

Recently, there has been much attention on improving the intuitiveness of ergonomics evaluation tools, as industrial operators should not be expected to have a background in ergonomics. Previous works have used digital human models (DHMs) alongside different types of visual cues for ergonomics evaluation: visualization of the human DHM with colored joints [14] or displaying relevant information such as COP [15], overloaded joint torque [11], level of fatigue [16].

The data relevant to assessing ergonomics is high-dimensional, including kinematic and dynamic state variables related to posture and efforts. This high-dimensional data is difficult to interpret, even for experts. Physiological sensors such as surface EMG or EKG often measure critical information for ergonomics assessment, but they require considerable post-processing and are difficult to interpret promptly. They are mostly used for post-experimental analysis [17], while few applications of online use in human-robot collaboration exist [16]. Also, the dimensionality of the data may become a bottleneck real-time computation and interpretation. For this reason, many works proposed to reduce the dimensionality of a data set, reducing it to a set of representative principal features [18]. The reduced human representation has been used to execute for instance activity recognition and prediction [19] or ergonomic optimization [20].

Here, we focus on providing ergonomics visualization tools that are intuitive for most users—experts or otherwise—by using state-of-the-art methods for dimensionality reduction. We present a novel and potentially easy-to-use visualization tool for ergonomics assessment: Latent Ergonomics Maps (LEM). To construct the LEM, we project a given ergonomics score onto a 2D latent space that represents the human posture. In our previous work, we proposed to use latent representations to capture the human movement/posture [21]. Here, we extend the use of latent maps to encode ergonomics evaluation scores.

In this article, we present two supportive tools for the evaluation of strenuous efforts and postures:Online visualization of joint angles and torques for ergonomic feedback. Given a human posture, the framework calculates the ergonomic assessment and an estimation of the human effort. This latter estimation is derived from inversion of the Lagrangian model using variables (e.g., Intertia, Coriolis) extracted from a simulated DHM. This enables to quickly verify a body posture, captured online using a motion capture suit. This visualization consists of a DHM with color-coded visual cues that express specific locations and joints of the body postures that are particularly non-ergonomic, further facilitating the ergonomics assessment.Latent Ergonomic Maps(LEMs) for immediate overall 2D visual feedback on RULA and RULA-based (RULA-continuous) scores. The algorithm uses a state-of-the-art method for dimensionality reduction and generative network, namely Variational Auto-Encoders (VAE). VAE allows us to encode high dimension postures and to sample and decode variations of the same postures. The latter allows creating a LEM by sampling the latent space, decoding the posture, and applying ergonomic assessment to the posture.

We demonstrate the feasibility of the proposed frameworks in two different scenarios: first, we demonstrate the feasibility of our visualization tool for typical industrial activities, such as walking, bending, and overhead manipulation; second, we showcase our tool during a physical human-robot interaction.

## 2. Related Work

### 2.1. Dimensionality Reduction for Human State Representation

DHMs typically have many degrees of freedom. A depiction of the human status, therefore, requires high-dimensional vectors, which in turn increase the computational cost of the ergonomics status evaluation [22]. To reduce the dimensionality of a data set, a solution is to extract a reduced set of representative principal features from the original data set [23]. This procedure is usually applied to reduce the amount of data necessary to train a model and to reduce the risk of overfitting during the training [24]. Recent works have applied this concept to compactly represent human motion. For instance, Mandery et al. [18] were able to reach high accuracy in a human motion classification task using only four features. Human models with a high number of degrees of freedom impact the time performance of motion analysis. To improve speed in the motion processing of the DHM data, recent works have used a representation of the human motion in a latent space [20,21]. Marin et al. [20] showed that the use of a latent space human representation improved the performance of their application. Ikemoto et al. [25] use principal components analysis (PCA) to reduce the dimensionality of the postures of a humanoid robot.

Dimensionality reduction techniques applied to human poses are also used in movement planning or prediction [26]. This type of approach uses latent space representations for selecting points that complete a given path (movement prediction) or for joining two extremes (movement planning). By inverting the relationship which maps the high-order dimensional to the latent space, it is possible to recover the poses having the latent space representations. For instance, variational autoencoders have been used to predict the next pose given the current one [27,28].

Among dimensionality reduction techniques, autoencoders (AEs) and variational-AEs (VAEs) have high reconstruction ability and produce compact latent spaces. For this reason, they have been widely used for reducing the dimensionality of the human state and for movement generation [19]. Dermy et al. [21] address the problem of predicting future human whole-body movements given prior observations. They map high-dimensionalality trajectories into a reduced latent space using AE. Then the prediction is based on a probabilistic description of the movement primitives in the latent space, which reduces the computational time for the prediction to occur. In our work, we proposed a method for visualizing the ergonomic scores in a latent space obtained using VAE.

### 2.2. Ergonomics Evaluation and Human-Robot Collaboration

To plan appropriate collaborative actions, collaborative robots need to have an estimation of the current human posture [20,29]. They must be able to calculate the physical, physiological, and/or cognitive state of the human to act accordingly. The perception of the human state relies on sensors that can be placed in the environment, embedded in the robot, or worn by the human. State-of-the-art motion capture techniques based on infrared cameras and reflective markers, or more recently on inertial technology, remain widely used to provide high-fidelity and high-frequency measurements of human kinematics [30]. The captured data are fitted to a DHM designed to have similar properties to the human operator (height, weight, and structure). Ergonomics scores typically rely on kinematics and dynamics information about the human’s movement, which are extracted from the DHM. There are two main types of DHMs: musculoskeletal models, they have many degrees of freedom, and allow the analysis of the human movement by simulating the muscular efforts [31]; rigid-body models, which are simplified models with fewer degrees of freedom, where the human is represented as made of rigid body links [32]. While musculoskeletal models can be very accurate on a biomechanical standpoint, less accurate rigid-body models are much faster to simulate. As such, they are better suited for real-time applications such as model-based prediction, control, and ergonomics assessment [32].

The need to reduce musculoskeletal disorders in the industry leads to extensive research into measures of ergonomics. Many ergonomics assessment tools use the kinematic of a given worker to evaluate, and score a given task execution [33]. For instance, RULA [6] and REBA [7] evaluate the upper-body and full-body posture by scoring how far the worker’s current joint angle positions are from a neutral and safe position. Some works have also used assessments typically used in robotics, such as manipulability measures that can be associated with the user’s operational comfort at executing the task [34,35].

Postural-based approaches, however, do not consider the dynamics properties of the human model nor any dynamics interactions with the environment. In many ergonomics scoring sheets, external forces and loads can be considered (e.g., in EAWS it is possible to account for manipulated weights), but a more general approach considering efforts due to physical interaction with loads and robots is required to inform the human. For this reason, many works in human-robot collaboration use DHMs to estimate the status of human dynamics. Latella et al. [13,36] estimate the balance, and the internal force distribution of the human during a human-robot collaboration using F/T sensors at the robot’s end-effector, force plates at the feet, and/or sensorized insoles. Peternel et al. [10] estimate the human joint overloading torque, a quantification of the effect an external load has on a given body joint. Maurice et al. [12] estimate the human body joint torques by solving an optimization problem where the objective function encodes the human posture and the optimization variables include torques and wrenches of the human model.

In this article, we present intuitive visualization tools to provide online feedback to industrial operators about the ergonomics of their movements. In the next sections, we present the methods and technologies used to build our tools (Section 3) and then discuss how to use them in two experimental scenarios inspired by industrial activities, where the human is executing different activities with challenging whole-body postures, with or without the assistance of a robot (Section 4).

## 3. Methods

The main goal of this paper is to develop intuitive visualization tools to provide online ergonomic feedback to industrial operators. The system is designed to be applicable to diverse industrial scenarios in which a human operator may execute non-ergonomic movements, either working alone, or in interaction with a robot. The intuitive visualization tools enable the human to easily visualize the risks of their posture even if they are not ergonomic experts. The human operator could, visualize the latent maps and the DHM along with ergonomics suggestions on a display during a training session, where time can be dedicated to analyzing and improving the gestures and the postures associated with a workstation. Looking at the display is hardly possible during the regular work because the display could divert the attention of the worker and increase the risk of accidents; however, the display can be used for post-hoc analysis immediately after a gesture if the production time allows for it. The visual feedback could be also displayed to the worker using augmented reality glasses [37]. Interestingly, the visual feedback can be used directly by the human worker, as well as by colleagues and ergonomics experts that evaluate the worker’s activity at a workstation.

Here, we project the ergonomic information into a 2-dimensional latent space that is learned to represent the human posture. In previous work, we used latent representations to capture the human movement/posture [21]. Moreover, we showed they could be used to predict future whole-body movements and optimize movements to improve ergonomics in specific contexts [20]. Here, the latent space is used to build a map that is used to display ergonomics information: we train the latent space to maximize the human posture representation and the ergonomics characterization of the space. The result is an intuitive visualization tool, called Latent Ergonomics Map (LEM), which provides a synthetic representation of the ergonomics of motions. The LEM is coupled with a real-time simulation of the human using a DHM that displays localized ergonomics information using colored spheres placed in relevant body parts. Both the LEM and the simulation are updated online, with the information coming from the human motion tracking system and eventually with the robot’s data if the human is working in interaction with a robot. The ergonomics visualization tools are shown in Figure 1. The only requirement of our system is that it requires a real-time human motion tracking system like wearable or environmental sensors. In this work, the human posture is tracked using an Xsens MVN motion tracking suit. Other devices could be used, but reviewing them is beyond the scope of this article.

### 3.1. Digital Human Model

Our DHM is a rigid multi-body system, similar to a humanoid robot, with anthropometrics properties of the human (height, weight) to mimic their kinematic and dynamics. It has 66 segments and is based on the Xsens MVN model (Figure 2). Using such a model, it is possible to reproduce the human movement as recorded by the Xsens MVN motion tracking system, a wearable set of 17 sensors distributed along the human body.

We modeled the human spherical joints collected by the motion capture suit as a series of 3 one-dimensional revolute joints, where each DoF is controlled by a single actuator. The resulting DHM posture is represented by the 66 joints: the links are modeled with geometric shapes (parallelepiped, cylinder, sphere) scaled with the human height. The dynamic properties (e.g., mass) are computed from anthropometric data available in the literature [32], assuming a homogeneous density of the links and scaling with the human body mass.

The posture of the DHM is calculated using a one-to-one mapping from the original human poses, which are measured in our case by a human skeleton or motion tracking system. The position of the floating base of the DHM is calculated using the position of the pelvis, which is provided by the motion tracking system.

The DHM enables the estimation of the human internal torques, as described in previous work [12]. However, here we do not use force plates to retrieve feet wrenches: instead, we approximate the wrench acting on the human feet with the weight force generated by the body. This approximation introduces an error in the joint torque estimation, but it is tolerable for our visualization purposes as long as the joint torques are not used in a control loop. This allows transporting easily the framework while having an issue with the movement quality, especially for upper-body movements.

The internal torques are calculated starting from the classical Lagrangian formulation:(1)$$M\left(\bar{q}\right)\ddot{\bar{q}}+C\left(\bar{q},\dot{\bar{q}}\right)\dot{\bar{q}}+G\left(\bar{q}\right)=S^{⊤}\tau +\sum_{i=1}^{n_{k}}J_{p_{i}}{\left(\bar{q}\right)}^{T}f_{i}$$
where $\bar{q}=\left(x_{F},q\right)$ is the DHM extended posture composed by the floating base pose ($x_{F}\in R^{6}$) and the joints angles ($q\in R^{66}$), M, C, G are respectively the inertia, the Coriolis and the gravity matrices, *S* is the actuation selection matrix due to the free-floating base and τ is the vector of the DHM joint torques. Concerning the computation of the external wrenches acting on the *i*-th link ($f_{i}$), we consider the force sensed by the robot through the torque sensors ($f_{RH}=J_{F}\left(q\right)\tau_{F,ext}$, where $J_{F}$ is the robot Franka’s Jacobian matrix and $\tau_{F,ext}$ are the robot external torques) and the gravitational forces acting on the feet. The external wrenches acting on the *i*th link are multiplied by the Jacobian matrix from the world frame to the *i*-th link frame ($J_{p_{i}}$). The choice of using only the gravitational forces acting on the feet helps to improve the speed of the algorithm and facilitates the deployment of our tools for ergonomics feedback to situations where only the human skeleton information is accessible. We solve the Equation (Equation 1) for τ.

### 3.2. Ergonomics Scores

To estimate the motion ergonomics, we use scores based on kinematics information (RULA, RULA-C, manipulability) and dynamics information (joint torques). The Rapid Upper Limb Assessment (RULA) tool [6] is often used by ergonomists to evaluate work activities involving upper-body motion. It consists of a score ranging from 1 to 7, calculated based on the joint positions (posture), the known force/load applied to the worker’s arm, and how many times the activity is repeated. The time evolution of RULA during a work activity is likely to have discontinuities and plateaus that make it inconvenient to use for motion optimization or continuous postural assessment. To alleviate this problem, we define a continuous version of RULA: RULA-C ($\varepsilon_{rc}\in R^{+}$). To compute RULA-C, we fit a second-degree polynomial function to calculate intermediate scores for the RULA joints. The joint scores for each limb are combined with weighted sums, where the weights are computed from linear regressions of the standard RULA tables. To account for the comfort of movement, especially in presence of physical interaction, we consider the arm manipulability: it provides information about the velocity and force production capacity of the limb endpoint in different configurations [10]. It provides a piece of complementary information about the human posture to RULA and RULA-C about the capacity of the human to produce forces.

Finally, we compute human joint torques using DHM, to account for the postural efforts and the effect of external wrenches acting on the human. To display the ergonomic assessments intuitively (especially targeting end-users that do not have a comprehensive background in ergonomics), we choose to visualize the local ergonomics scores in color-coded spheres attached to relevant body parts in the DHM, updated at each time step. The mapping from the ergonomic score to the colors follows existing recommendations when available: green is associated with low ergonomics scores and indicates low risks, red with high scores indicating high risks, and yellow with scores in between the two. Each ergonomic score (kinematic and dynamic) can be normalized between zero and one by using the maximum value for the specific task, computed in an offline calibration phase. Normalizing the values enables to highlight for a specific task the differences in the ergonomics while executing the same movement in different ways. We are aware that this normalization has some limitations (e.g., encountering new movements that are particularly risky). In future work, we want to overcome these limitations and integrate activity recognition [38] into our framework, to change normalization values according to the current activity and movement.

### 3.3. Latent Ergonomics Maps (LEMs)

Latent Ergonomics Maps (LEMs) project an ergonomics score on a 2-dimensional latent space that is trained to represent the human posture (note that this means that each ergonomics score listed in Section 3.2 leads to a different map). Previous work showed the advantages of reducing the dimensionality of the human postural state (Section 2.1). In particular, Dermy et al. [21] showed that low-dimensional (2 to 7) latent representations of a human posture are sufficient for probabilistic models computed over the latent space to predict future whole-body movements with accuracy for activity recognition and postural evaluation. Malaisé [22] showed that a 2D latent space is sufficient for activity prediction and to some extent for postural evaluation. In this article, we propose a synthetic representation of the human postures using a 2D latent space created by a Variational Auto-Encoder, and then we project ergonomics scores of sampled human postures on the latent space, in the form of a map. The resulting map is the Latent Ergonomics Map. This representation allows us to visualize in an immediate way when a person is in a non-ergonomic (associated with red color) or ergonomic (associated with green color) posture. In future work, we would like to use the presented space to provide corrections to humans and plan better ergonomically optimized trajectories.

The LEM is created in an offline phase, requiring a dataset of human movements and the definition of the ergonomics score. Training a LEM proceeds in two steps. The first step is to create and train a Variational Auto-Encoder (VAE) to represent the human postures. A schematic of the training procedure is shown Figure 3. A VAE is an auto-encoder based on variational inference [39]. Let ${\left[x_{k}\right]}_{k=1}^{K}$ be a data-set of *K* independent and identically distributed samples of some continuous observation variable *x* of unknown distribution. It is assumed that *x* is generated by some process involving the latent variable *z* and the parametric functions of distribution $p_{\theta^{*}}\left(x|z\right)$ and $p_{\theta^{*}}\left(z\right)$: $x∼\int_{z}p_{\theta^{*}}\left(x|z\right)p_{\theta^{*}}\left(z\right)dz$ where $\theta^{*}$ is a set of parameters. In a VAE, the so-called *decoder* neural network tries to fit the function which maps *z* to *x*, and so learns from data the values of the weights θ. The distribution is assumed to be Gaussian: $p_{\theta }\left(z\right)=N\left(0,I\right)$ and $p_{\theta }\left(x|z\right)=N\left(\mu_{x},\sigma_{x}^{2}\right)$. However, the transition function $p_{\theta }\left(z|x\right)$ is not known. A recognition model $q_{\phi }\left(z|x\right)$ is used to approximate true posterior $p_{\theta }\left(z|x\right)$. In our case the distribution is also assumed to be Gaussian for simplicity, but without loss of generality: $q_{\phi }\left(z|x\right)=N\left(\mu_{z},\sigma_{z}^{2}\right)$, where ϕ is represented by weights and biases of a neural network (*encoder*). Training the VAE aims at recovering values of the parameters (θ,ϕ) in such a way to approximate as much as possible the optimal parameters $\left(\theta^{*},\phi^{*}\right)$.

We choose a relational VAE to define a loss function which represents both the reconstruction loss and a loss on the relation reconstruction [40]:(2)$$L=\left(1-\alpha \right)D_{kl}(q_{\phi }\left(z|x\right)||p_{\theta }\left(x|z\right))+\alpha D_{kl}(q_{\phi }\left(z|\varepsilon \left(x\right)\right)||p_{\theta }\left(\varepsilon \left(x\right)|z\right))$$
where $D_{kl}\left(.\right)$ is the Kullback-Leibler divergence, ε(.) is the relation function (in our work corresponding to the ergonomic loss function RULA-C) and, finally, the parameter α is a scale parameter to control the weights of the data reconstruction loss and the relationship reconstruction loss. This enables us to obtain a latent space that not only represents the postures, but also, tries to encode the space to minimize the error in reconstructing the correct ergonomics score for the compressed posture.

Training the VAE requires a dataset of human movements. For example, the AnDy dataset [30] contains many examples of human activities, recorded with the Xsens motion tracking suit, which can provide both wearable sensing data and estimated postural data. To simplify, let us assume that we input human postures fitted to the DHM ($R^{66}$). We train the VAE with batches of human postures from the training dataset to find the parameters of the encoder-decoder networks, to minimize the loss function Equation (Equation 2) via back-propagation.

Once the VAE is trained, the second step consists in creating the ergonomics landscape projected on the latent map. The procedure is illustrated in Figure 4. We uniformly sample in the latent space a set of 2D vectors ($z_{i,j}=\left[z_{i},z_{j}\right]\in R^{2}$). We reconstruct the samples using the *decoder*. Then, for each reconstructed human posture (${\hat{q}}_{i,j}\in R^{66}$) we calculate the ergonomic score $\varepsilon ({\hat{q}}_{i,j})$ (e.g., RULA or RULA-C) from the estimated human joint angles, following the method presented in Section 2.2. Note that one map is created for each ergonomics score (for instance, one map for the standard RULA score, and another map for the RULA-C score). If the data used for the training are subject to a normalization function $\left(f_{n}\left(.\right)\right)$, it is necessary to invert the normalization $\left(\varepsilon (f_{n}^{-1}\left({\hat{q}}_{i,j}\right))\right)$ before calculating the ergonomic scores. The result is a height-map composed by the 2D coordinated in the latent space and the ergonomic score ($\left[z_{i},z_{j},\varepsilon \left({\hat{q}}_{i,j}\right)\right]\in R^{3}$).

The LEM can be used online to display the movement that the human operator is executing, which appears as a trace on the landscape: since every point is a posture, the height of each point indicates the ergonomics evaluation associated with the corresponding human posture. The human posture is recorded with wearable sensors (e.g., in our case the Xsens MVN motion tracking suit) and retargeted into the DHM ($q_{c}\in R^{66}$). The *encoder* neural network reduces the input to its 2D representation ($z_{i,j}$), the ergonomics score is then retrieved. We adopt a triadic color code to make the height map more intuitive to read: we associate low ergonomic scores (e.g., ergonomic postures, with low risk) with green; high ergonomic scores (e.g., non-ergonomic postures, with high risk) are associated with red; middle scores are associated with yellow. This kind of representation allows to visualize in an immediate way when a person is in a non-ergonomic (and therefore associated with red color) or ergonomic (associated with green color) posture space. Similarly, different variations of the same movement (and therefore different postures) can be mapped in the latent space, as it was done in Figure 5, to visualize which trajectory is better from an ergonomic point of view. In future work, we would like to use this representation to provide corrections to humans and plan better ergonomically optimized trajectories.

## 4. Experiments

In this section we elaborate on the structure of the VAE and the specific training procedure for the experiments. We then describe the use of the trained LEM to provide online ergonomics feedback in two different experiments, where a human executes pick and place activities with and without robot assistance.

### 4.1. Setup and Scenarios: Experiment 1

We demonstrate our ergonomics visualization tools to evaluate human movements executed in two experiments, with two work-related scenarios. In the first experiment, the human performs a pick and place task. The task is inspired by packaging tasks on assembly lines in the manufacturing industry and consists of picking, carrying, and placing a 6 kg bar. One male participant performed 8 sequences of the task, with each sequence consisting of 6 to 8 pick-and-place actions. Each sequence started and ended in the same neutral pose. The bar was initially placed at a height of 45 cm on a 100 × 50 cm flat support. The participant was instructed to take the bar with both hands, carry it to the other side of the support, place the bar there and return to the initial position to perform the next iteration. Each sequence lasted around one minute. To add variability to the data, the participant was instructed to change the position of his hands on the bar and to follow two different paths when going to and coming from the bar’s final position.

### 4.2. Setup and Scenarios: Experiment 2

In the second experiment, the human carries loads in collaboration with a robot. They move a box together from a point *A* to a point *B* and then backward, along a trajectory that lasts about one minute. The box is initially placed at a height of 85 cm on a table. The box is fixed to the robot end-effector which carries the majority of its weight; the human grasps the object through a handle as shown in Figure 6. Each sequence lasts around one minute. At the beginning of the activity, the robot and the human grasp the object, and each sequence starts and ends in the same neutral pose. The robot control is based on a Cartesian impedance controller:(3)$$\tau =J{\left(q\right)}^{⊤}\left(D\left({\dot{x}}_{d}-\dot{x}\right)+K\left(x_{d}-x\right)\right)+g\left(q\right)$$
where $K\in R^{6\times 6}$ and $D\in R^{6\times 6}$ are respectively the stiffness and the damping matrices and g(.) is the gravity compensation term. Cartesian impedance control generates a torque proportional to the error between the end-effector pose and the desired end-effector pose ($x,x_{d}\in R^{6}$) and their derivatives (${\dot{x}}_{d},\dot{x}\in R^{6}$).

We implemented two Cartesian impedance behaviors changing the values of the stiffness matrix: in the first, the robot was more compliant (K=500N/m) and in the second the robot was stiff (K=1000N/m). The desired Cartesian damping *D* was calculated proportional to *K* using factorization design as in [41]. The robot’s trajectory is predefined by the robot planner, the latter selects a trajectory of Cartesian points using a parabolic curve passing through the initial and final points while the orientation is maintained constant. A participant repeated the movements 15 times, with 3 different robot trajectories (5 times for each trajectory) to add variability to the movement; in particular, we designed different robot trajectories in such a way to induce a variety of human postures and ergonomics. Some movement examples are shown in the video attachment. The robot used is the Panda (Franka Emika, Munich, Germany) robot controlled using *libfranka* and *FrankaROS* libraries.

In both experiments, the human is equipped with an Xsens MVN motion tracking suit, which provides real-time (240 Hz) information about the current human pose: joint angles and link’s positions, orientations, velocities, and accelerations. Both the human’s and the robot’s sensors measurements are streamed online to the module, to visualize and simulate their movements in the simulation (Dart [42]) as shown in Figure 7.

An overview of the main modules used in the two scenarios is shown in Figure 8, which details as well the flow of information from the Xsens sensors and the Franka robot. The simulation receives the sensed information, from both humans and robots, and their corresponding models are updated. All the items in the scene are modeled (i.e., a Unified Robotic Description Format (URDF) is available for all the elements: robot, experimental setup—table, object, etc.—as well as human) and simulated. Precisely, the robot communicates the following information to the simulation: (1) Robot joint configuration, (2) Robot end-effector pose, (3) External measured torques, and (4) Internal torques. The robot pose (joint angles, end-effector pose, and torque) are used to update the simulated robot. Meanwhile, the measured external torques are used to calculate the wrenches that the human is applying to the robot and then used to calculate the effort estimation Equation (Equation 1) Similarly, the motion tracking suit communicates: (1) the human posture expressed as the joint angles, and (2) the positions of the human links. The human joint angles are used to generate the LEM and to simulate the DHM; the links’ poses are used to locate the DHM in the reference frame of the simulation.

The main interest of the second scenario is to use a DHM with visual clues of the joint torques and efforts, to account for the physical interaction between the human and the robot. The contact wrenches are input in the DHM in real-time, using the Franka robot’s sensor measurements, which enables a better representation of the ergonomics status of the human from the point of view of efforts.

### 4.3. Creating and Visualizing the LEMs

To obtain the LEMs for every ergonomics score, we first created and trained the VAE representing the human body postures. The VAE architecture consists of 5 layers with 66 inputs, 200 hidden neurons, 2-dimensional latent space, and 66 outputs, where rectifier, *tanh* and identity activations are used for the hidden layers and the output layer. The hyperparameters of the VAE are chosen based on the reconstruction error and the training time by grid search. To train the latent space, we perform first a pre-training with a large human motion dataset (*AndyData-lab-onePerson dataset* [30], containing more than 5 h of recorded data), then a fine-tuning with smaller task-specific datasets acquired for each experiment (see below). This latter step is optional. The first training with a large dataset is sufficient to generate a LEM, in particular, it enables to obtain a generic LEM that is task-independent and that captures a variety of different postures. In our experiments, however, the fine-tuning enables us to obtain a task-specific LEM that is more specialized to represent the postures of a specific task, while maintaining the generalization capabilities. All the datasets were normalized between zero and one and augmented using intermediate and mirrored poses before training.

The VAE training was carried out according to the procedure explained in Section 3.3. During training, we optimized both the weight of the *encoder* and the *decoder*. The weight optimization was done on an Intel Core${}^{TM}$ i7-8850H with 6 cores at 2.6 GHz, requiring about 20 min. The trained *encoder* can be used to visualize example movements in the latent space. This can be useful for preliminary insights into the postures for different activities. For example, Figure 9 shows the sequence of postures, projected in the 2D latent space, associated with the movements from the dataset of [38], which contains activities such as bending, kicking, lifting objects, and walking. Some postures are also displayed, to clarify that each 2D point is representing a different posture. Some 2D points belong to different activities, and this is normal since the same body posture can be observed in different activities. It is also possible to display ergonomics information in this activities representation: as shown in Figure 5, it is possible to color-code the points associated with a movement according to the ergonomics scores. In this case, the activity information is lost, but it is possible to inspect one or more movements. Of course, the limit of this visualization is that it only shows the execution of a movement.

The trained *decoder* is used to create the LEM, which is a latent space height-map for a given ergonomics score. To do so, we uniformly sample the latent space to obtain a set of 10,000 points (100 × 100). For each point in the latent space we apply the *decoder* to reconstruct the original posture, we invert the normalization function and then we apply the ergonomic score to the estimated pose. The shape of the latent space is influenced by the choice of the activation functions, for instance, the sigmoid activation function maps the input to [0,1] while the tanh maps to [−1,1]. In Figure 10, we show the LEMs for RULA and RULA-C. The triadic color code (green, yellow, red) is applied to the postures classified according to the RULA recommendations, while a continuous color coding based on the triad is used for RULA-C. The LEM created using the RULA-C score creates, by definition, a latent space that is similar to that of the RULA but continuous. While there is no immediate advantage in using one of the two LEMs for ergonomics visual feedback, the continuous LEM can be used in future work as a reduced model for planning ergonomics movements using gradient-based methods [43].

In the online phase, the LEM is used to visualize the overall ergonomics score of a posture. Figure 1 shows the RULA LEM (center) corresponding to the human motion (left) tracked in real-time (240 Hz) by the Xsens MVN suit. During the execution of the movement, the current human posture is projected into the ergonomic map, after being encoded by the *encoder* into a 2D point in the latent space. The result is a trace, i.e., a sequence of 2D points moving in the ergonomic map, representing the human motion and its ergonomics evaluation. Another example of the trace onto a LEM is shown in Figure 6, and other examples are available in the video attachment.

### 4.4. Visualization of Local Ergonomics Scores in the DHM

The DHM is used to display in real-time the ergonomics information associated with human movement. The real-time information about the human posture is used to update the DHM pose and compute the local ergonomics score, where “local” indicates the specific body location. Each score is calculated using the partial scores presented in the RULA table. In the case of RULA-C, these have continuous values because they are the result of polynomials that approximate the RULA values. The local ergonomic scores are then visualized on the DHM to provide intuitive feedback about the current posture. Specifically, we color some spheres placed at relevant joints (knees, ankles, back, torso, shoulders, and elbows) with colors that are proportional to the ergonomic score. The latter is normalized in the range [0,1]→ [GREEN, RED], using the maximum score value that is set after the maximum score registered for that body part on the dataset used for training the LEM in the offline phase. The reason for the latter normalization is to better spot the difference in ergonomic scores between similar movements. In our setup, it is possible to select the ergonomic score to visualize on the DHM by interacting with an easy-to-use GUI (some examples in the video attachment). Figure 1 shows the DHM displayed during a manipulation task, together with the RULA LEM: while the LEM provides a synthetic visualization of the overall ergonomics score, the spheres on the DHM enable to visualize the individual scores on the different body parts. This is convenient to identify the body parts that are more at risk, from the ergonomics standpoint, which are colored in red; those that are not at risk are colored in green, and the intermediate values in yellow. Figure 11 shows more examples of the DHM during a pick and place movement where the human takes on different body postures: back joints are red when the human is bent forward, the knees are red during squats, while shoulders are red during over-head work.

## 5. Discussions and Conclusions

In this article, we presented a set of tools for providing online ergonomics feedback to human workers during their activities, also when they physically interact with robots. A Digital Human Model is used to visualize, with color-coded spheres, the body areas and joints that are subject to efforts and non-ergonomic postures according to state-of-the-art ergonomics scores, such as RULA. Several plotting tools were also developed to provide an intuitive visualization of ergonomics scores associated with human movement.

Our contribution, Latent Ergonomics Maps, are synthetic representations of the overall ergonomics scores projected onto a bidimensional latent space that maps human postures. The result is an intuitive color-coded map where the human posture is a point, a movement is a line, and their associated color is an estimation of the ergonomics score of choice. LEMs can be used for bio-feedback or self-correction, as a visual tool for teaching, or simply to inform the human. Their potential goes beyond the online feedback for the human, as they can be used to inform the robot as well, which can find applications in planning ergonomically optimal collaborative motions. Advantages of LEMs include their ease of interpretation also for non-experts, and the computational efficiency, enabling online feedback. A limitation of projecting the map on a 2D latent space, which is necessary for visualization purposes, is the information loss that may result from such a strong dimensionality reduction. However, the error is tolerable for ergonomics scores based on postural information, and otherwise acceptable if coupled with the visualization of efforts on the Digital Human Model.

In the future, we plan to combine the prediction of intended movement [21] with LEMs, therefore predicting future ergonomics scores for the intended movement. This will enable us to alert the human of possible risks associated with ergonomics. Moreover, we want to leverage the LEM for ergonomics optimization of robot motions to improve human-robot collaboration. The idea is to inform the robot about the ergonomics risk associated with planned collaborative robot trajectories as in [44], and then to optimize these trajectories using ergonomics optimization as in [43,45].

## Figures and Tables

**Figure 1 sensors-22-03981-f001:**
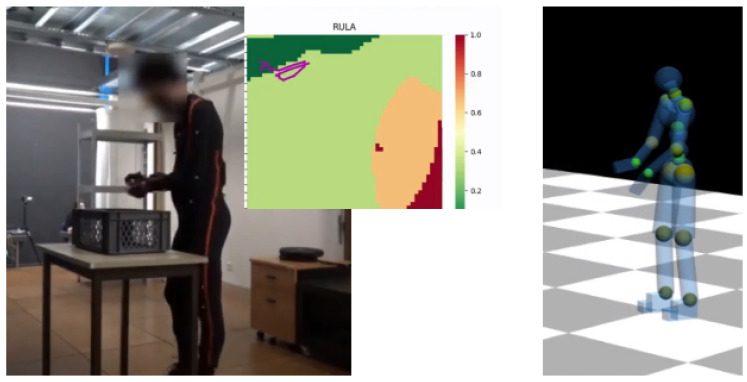
The ergonomics visual feedback tools used to measure the executed motion in real-time. (**Left**): the human movement is tracked with the Xsens MVN motion tracking suit. At the same time the RULA Latent Ergonomics Map is calculated, with the visualization of the current human movement (the magenta-colored line). (**Right**): the DHM with colored spheres showing the RULA scores at relevant body locations.

**Figure 2 sensors-22-03981-f002:**
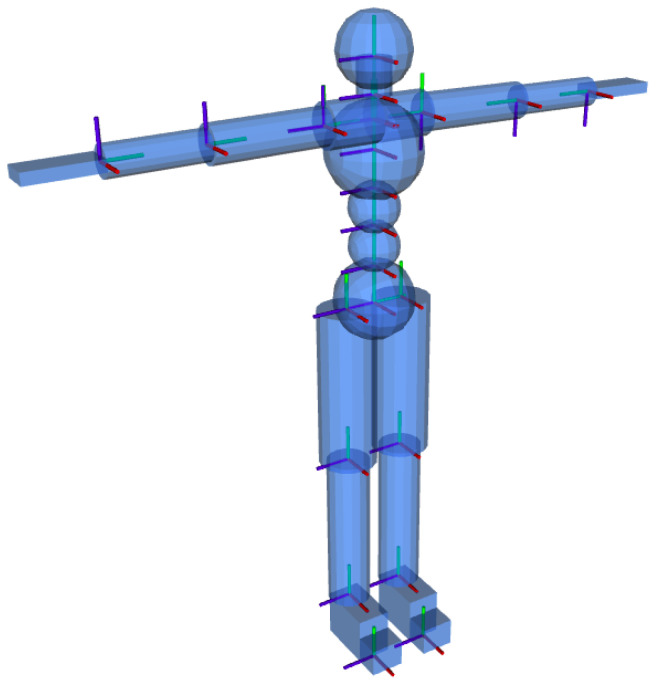
Digital Human Model (DHM) with 66 degrees of freedom. For each DHM’s link we display its origin axis (*x* in red, *y* in blue and *z* in green).

**Figure 3 sensors-22-03981-f003:**
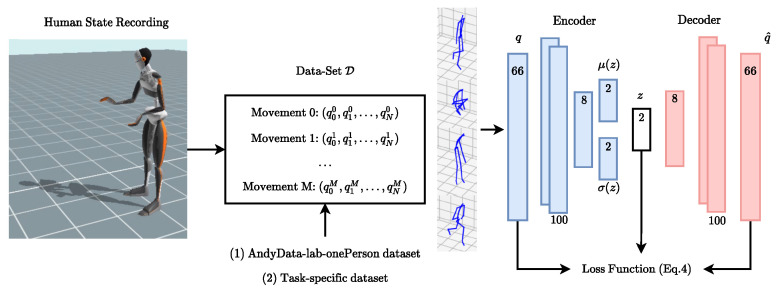
Training the VAE of the LEM requires a dataset of human postures. For each posture an ergonomics score exists. In our experiments, as a dataset, we used: (1) the AnDy dataset [30] and (2) a task-specific dataset acquired for each experiment. The training is done to find the VAE weights which simultaneously minimize the reconstruction error of the postures and the associated ergonomics score.

**Figure 4 sensors-22-03981-f004:**
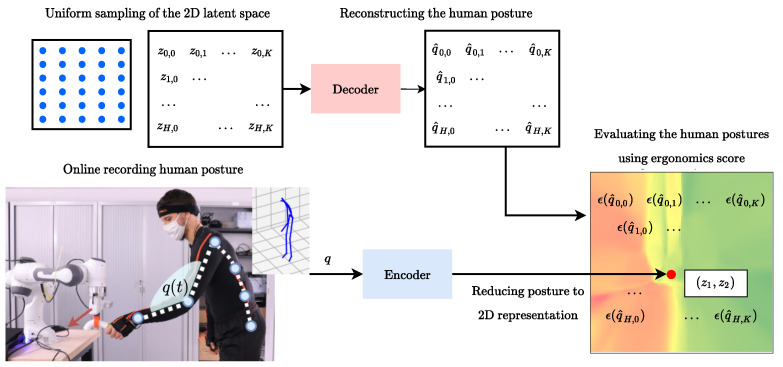
Creation of the LEM: the 2D latent space of the VAE is sampled. The *decoder* reconstructs sampled latent points. The reconstructed postures are used to compute the ergonomics scores. The ergonomics score is therefore associated with the original 2D points, thus creating a heightmap.

**Figure 5 sensors-22-03981-f005:**
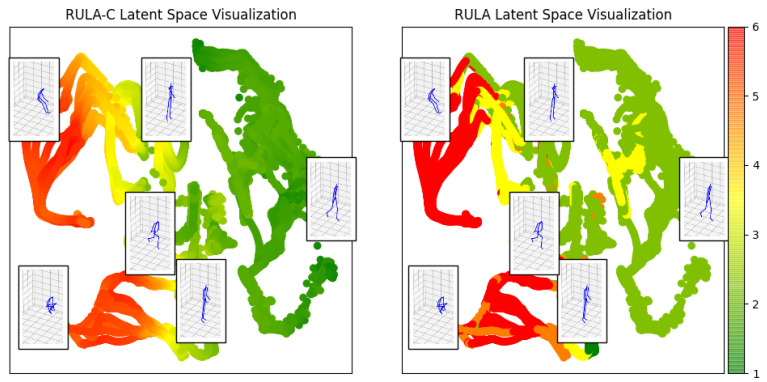
Latent space representation: in this plot, some examples of movements from the dataset of [38] colored according to the ergonomic evaluation. The postures associated with some 2D points in the latent space are shown. On the right, the scale of the colors corresponds to the ergonomic scores where: 1 (green) corresponds to the safer postures while 7 (red) represents the postures less safe.

**Figure 6 sensors-22-03981-f006:**
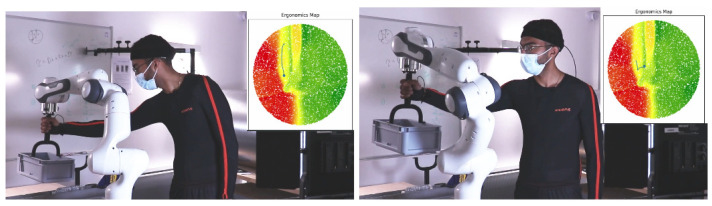
Experiment 2: collaborative object transportation. The human is physically interacting with the Franka robot. On the right of each photo, the Latent Ergonomic Map. The current human posture is a point on the map while the line that is attached to it represents the previous human posture.

**Figure 7 sensors-22-03981-f007:**
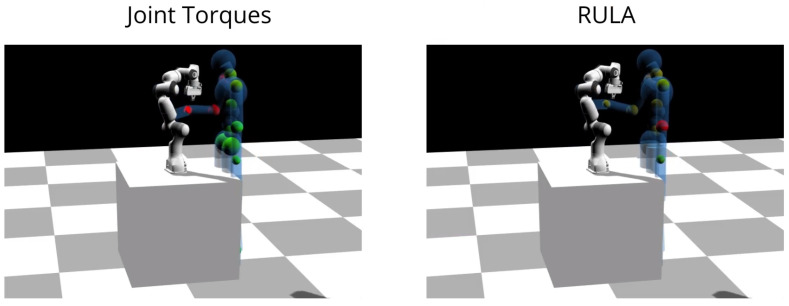
Experiment 2: collaborative object transportation. The DHM with the colored spheres indicating non-ergonomic joint values.

**Figure 8 sensors-22-03981-f008:**
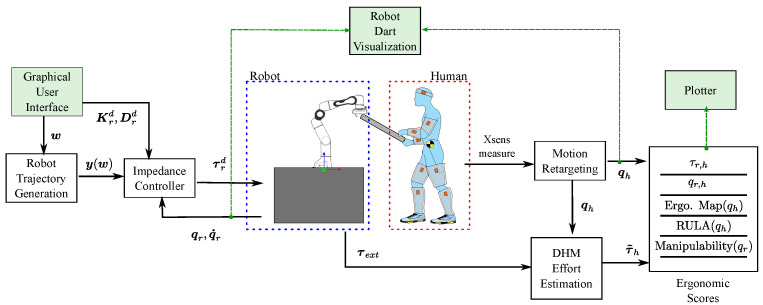
Overview of the system in a human-robot collaboration setting. In the left box: the robot modules that enable to control the robot and retrieve information about the contact forces exchanged with the human. The human-robot interaction force is measured on-line thanks to the joint torque sensors embedded in the robot. Each of the 7 axis is equipped with a torque sensor, and based on these torques measurements, the Franka API provides an estimation of the interaction force at the end-effector. In the right box: modules for online estimation of the human kinematics, dynamics, ergonomics scores and visualization tools. Green boxes: the framework includes visualization tools to plot the online ergonomics scores and other relevant quantities, as well as visualizing the human and the robot interacting in a digital twin based on Dart physics engine.

**Figure 9 sensors-22-03981-f009:**
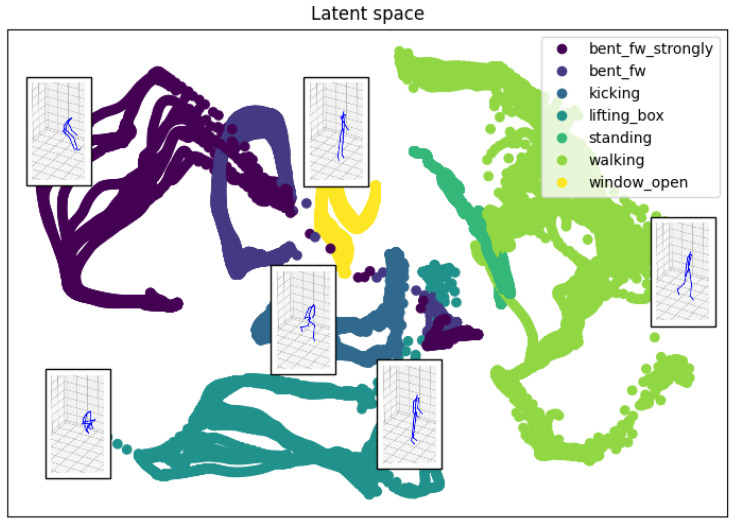
Latent space representation of the human movement during some activities from the dataset of [38]. The movements presented in this latent space are: bent forward (strongly), kicking, lifting a box, standing, walking, open a window. A movement is a sequence of points in the latent space. A color code enables to distinguish the 2D path associated with each sequence. The postures associated with some 2D points in the latent space are shown.

**Figure 10 sensors-22-03981-f010:**
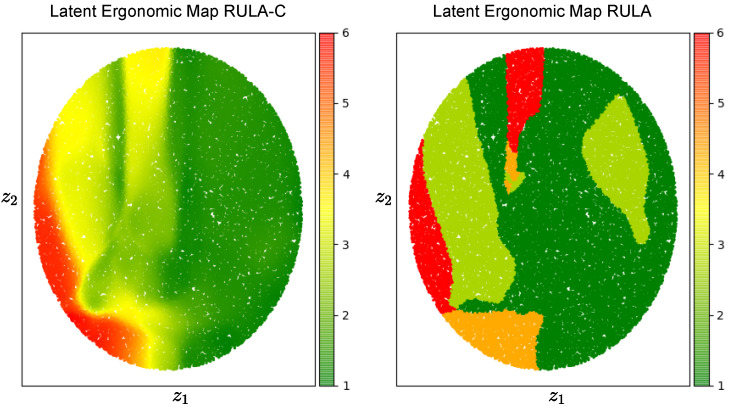
Latent Ergonomics Maps (LEMs): one for RULA, one for RULA-C. The figure visualizes the ergonomic data generated by the *decoder* network of a variational autoencoder. Here, we’ve sampled a grid of values around the origin with a radius of size 1 from a two-dimensional Gaussian and displayed the output of our *decoder* network. The distinct ergonomic scores which exist in different regions of the latent space smoothly transform from one to another.

**Figure 11 sensors-22-03981-f011:**
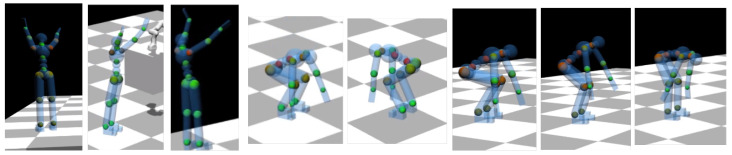
Experiment 1: pick and place. Across the pick and place task, the human takes on different postures. In particular, some are usually classified as non-ergonomic (e.g., hands over shoulders, bent back, squat). The color-coded spheres on the DHM show the body parts that have a high risk (ergonomics score: RULA) during these postures.

## Data Availability

The dataset of human motions and postures is available at https://zenodo.org/record/3254403, accessed on 18 May 2022. A detailed description of the dataset and its content is presented in [30].

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
