# Peer review of "Latent Ergonomics Maps: Real-Time Visualization of Estimated Ergonomics of Human Movements"

_sensors, 2022, doi:10.3390/s22113981_

Round 1

Reviewer 1 Report

Overall, this is a solid paper with a substantial amount of work involved. The paper should be improved from the following aspects:

  1. The title of paper, the first letter of each word should be capitalized.
  2. The writing of the paper is not that good, the authors should invite professors or professional institutes to polish the writing.
  3. In the related work, relevant and high influence papers are  missing, including but not limited to: 
    a. Investigating Pose Representations and Motion Contexts Modeling for 3D Motion Prediction, TPAMI 2022, doi={10.1109/TPAMI.2021.3139918}
  4. Rectangular boxes in Figures 3-5 should be redrawn to make them look more appealing.

Author Response

Thanking the reviewer for suggested corrections, we attach the cover letter containing responses to the reviewer's suggestions. We believe that with your help we have improved the quality of the paper and made it clearer.

Reviewer 2 Report

The paper presents a set of tools to visualize status of human motion ergonomics in human-robot collaboration or other works. The key contribution lies in the proposal of the so-called “latent ergonomics maps” based on the variational autoencoder (VAE) dimension reduction. While the manuscript is in general well-written, the reviewer has the following concerns to be resolved before it can be recommended for publication.

  • Line 152, the grammar used here is weird, should be "an … tool is presented to …."
  • Line 163, how is the risk visualized to a human in work?? Should the worker wear some sort of AR glasses?
  • Line 184, does this 66 degree of freedom mean that 66 sensors will be installed on corresponding position of a real human to collect the data? A vector of 66 dimensions does not seems a large one, which can be easily processed and visualized in a short time.Why bother to reduce dimensions?
  • Line 211, typos here, remove a "the"
  • Line 288, How is this score calculated? What's its relationship with the encoded 2D vector?
  • Line 316 to 317, the figure number should appear in sequence as far as possible. Here, the authors refer to fig. 10, which is the last figure of this paper without figures in the middle cited in between.
  • Figure 5, there is no information exchange between human and robots? Then how can the robot "retrieve information about the contact fores exchanged with the human"?
  • Line 416, should be 10,000
  • Line 439, does this local score correspond to the different color and size of spheres in fig. 6? How is it different form the proposed LEM visualization method?

Author Response

(The authors gave the same response as above.)

Round 2

Reviewer 2 Report

The authors addressed my comments properly, and thus it is recommended to accept the paper.